# Few-Case Fine-Tuning Overcomes Domain Shift in Mitosis Detection and Atypical Classification Across Cancer Types

**Mieko Ochi**        MIOCH@NCC.GO.JP and **Yusuke Yamamoto**        YUYAMAMO@NCC.GO.JP
*Laboratory of Integrative Oncology, National Cancer Center Japan Research Institute, Tokyo, Japan*

**Yukako Yagi**                                                                YAGIY@MSKCC.ORG
*Department of Pathology and Lab Medicine, Memorial Sloan Kettering Cancer Center, NY, USA*

## Abstract

Mitotic figure counting is essential for tumor grading but shows high inter-observer variability, while automated typical/atypical classification—carrying independent prognostic significance—remains largely unexplored. Deep learning models pretrained on large multi-domain datasets still suffer domain shift when applied to new institutional settings—even when the target cancer type is represented in the pretraining data. We show that parameter-efficient fine-tuning on only five whole-slide images (WSI) per cancer type overcomes this institutional shift: on breast invasive ductal carcinoma (IDC) and pancreatic neuroendocrine tumors (PanNET)—both present in the pretraining corpus yet acquired at a different institution and scanner—fine-tuned models surpass the pretrained baseline and pathologists in detection, while a foundation-model ensemble achieves balanced accuracy 0.81–0.82 ($\kappa$=0.59–0.63) in typical/atypical classification, exceeding all pathologists. We further present a human-in-the-loop (HITL) WSI viewer for iterative model refinement through automated adapter-based fine-tuning.

**Keywords:** mitosis detection, atypical mitosis classification, domain adaptation, human-in-the-loop, computational pathology

## 1. Introduction

Mitotic figure counting is integral to tumor grading (Elston and Ellis, 1991; Rindi et al., 2022), and atypical mitoses carry independent prognostic significance (Balkenhol et al., 2022), yet manual identification shows high inter-observer variability (Yousif et al., 2022). Pretraining on MIDOG++ (Aubreville et al., 2023) (503 cases, seven tumor types) has advanced automated detection (Aubreville et al., 2024), yet pretrained models face *domain shift* at new institutions even when the target tumor type is in the pretraining data.

We show that **(1)** fine-tuning a pretrained detector (Lv et al., 2025) on only five WSIs (0.4% of parameters) overcomes institutional shift on breast IDC and PanNET—both in the pretraining corpus—surpassing pathologists; **(2)** a foundation-model ensemble reaches BA $= 0.81$–$0.82$ ($\kappa = 0.59$–$0.63$) in typical/atypical classification, exceeding all pathologists ($\kappa \approx 0.24$–$0.25$); and **(3)** a HITL WSI viewer enables iterative refinement via automatic adapter fine-tuning.

## 2. Methods

**Datasets.** We evaluate on breast IDC and PanNET (5 train + 5 test WSIs each, MSKCC; scanners: Hamamatsu NanoZoomer S60 [PanNET], 3DHISTECH Pannoramic 1000/confocal [breast]). PanNET GT used PHH3 IHC registered to H&E (328 test mitoses); breast GT used two-pathologist H&E consensus (163 test mitoses). Two independent pathologists annotated test slides per type.

**Detection models.** KongNet (Lv et al., 2025) is an Attention U-Net with EfficientNetV2-L encoder pretrained on MIDOG++ (Aubreville et al., 2023). Three variants: *Baseline* (pretrained only); *Decoder* (encoder frozen, decoder + heads trained; 7.9% of parameters); *Adapter* (encoder and decoder frozen, with lightweight bottleneck adapters—Conv-BN-GELU-Conv with residual, reduction 4—inserted after each encoder stage; 0.4%). For each WSI, a $3\,mm^2$ region around annotated mitoses is used for training and inference. Fine-tuning uses LOSO 5-fold cross-validation with AdamW (decoder lr=$10^{-4}$, adapter lr=$5\times10^{-4}$, weight decay $10^{-2}/10^{-3}$), a Jaccard + Dice + Focal loss, a positive-tile-balanced sampler (target 30% positive tiles per epoch), and early stopping on validation nucleus F1 (patience 8). Final predictions are a hybrid ensemble averaging the held-out fold model with two MIDOG++ checkpoints.

**Classification models.** Two models are fine-tuned on target data via LOSO 5-fold CV: (a) *MIDOG2025 Ensemble* (Ochi and Yuan, 2025)—four foundation models (UNI (Chen et al., 2024), Virchow (Vorontsov et al., 2024), Virchow2, ConvNeXt V2) with LoRA (r=4, $\alpha$=4) on the ViT backbones and full fine-tuning for ConvNeXt V2, trained with PolyLoss + Adam under cosine annealing, with 8-direction ($D_4$: H/V flip × transpose) TTA at inference; (b) *ChroMa-GI* (Banerjee et al., 2025)—DDPM inpainting-based augmentation ($\sim$26k synthetic crops added to the AMi-Br training set) with an EfficientNet-B0 classifier (Adam, lr=$10^{-3}$, BCE), 5-fold CV ensemble.

**Inference.** WSI detection uses overlapping $512^2$ px tiles (stride 256) at $0.25\,\mu m$/px (native: PanNET $\sim$0.23, breast $\sim$0.16), $4\times90°$ TTA, and max-merge maps; centroids matched to GT by greedy distance (15 px).

**HITL viewer.** A web application (FastAPI + OpenSeadragon, Figure 1) supports interactive ROI inference, manual annotation, and GeoJSON export; new annotations trigger automatic adapter fine-tuning, with model update on improvement (Budd et al., 2021).

## 3. Results

Table 1 summarizes performance across both cancer types.

**Detection.** On breast IDC the Baseline (F1 = 0.461) barely matches the best pathologist (0.459), illustrating domain shift; Adapter fine-tuning raises F1 to 0.509 (+0.048), surpassing all pathologists. On PanNET, Baseline 0.629 / Adapter 0.641 both exceed the best pathologist (0.383) by $\geq$ 65%. Inter-pathologist F1: 0.425/0.405.

**Classification.** The MIDOG2025 Ensemble gives the highest BA on both types (breast IDC 0.807, $\kappa$=0.59; PanNET 0.822, $\kappa$=0.63), beating every pathologist and ChroMa-GI (0.760/0.684); inter-pathologist $\kappa$ was only 0.239/0.252.

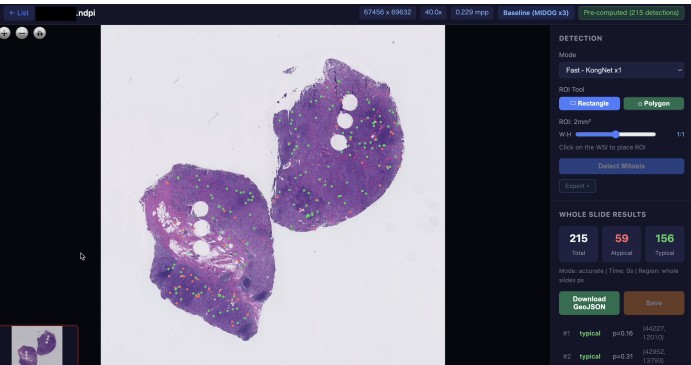

Figure 1: HITL WSI viewer with detection results, ROI inference, and annotation tools.

Table 1: Detection F1 and classification BA $(\kappa)$. PT: pretrained on MIDOG++. FT: fine-tuned on 5 WSIs. [†]F1 for detection, $\kappa$ for classification.

| Method | Detection F1 | | Class. BA $(\kappa)$ | |
|---|---|---|---|---|
| | Breast IDC | PanNET | Breast IDC | PanNET |
| KongNet Adapter (FT) | **0.509** | **0.641** | – | – |
| KongNet Decoder (FT) | 0.488 | 0.637 | – | – |
| KongNet Baseline (PT) | 0.461 | 0.629 | – | – |
| MIDOG25 Ens. (FT) | – | – | **.807** (.59) | **.822** (.63) |
| ChroMa-GI (FT) | – | – | .760 (.49) | .684 (.45) |
| Pathologists $(n{=}2)$ | .415–.459 | .372–.383 | .661–.781 | .704–.748 |
| Inter-path.[†] | 0.425 | 0.405 | 0.239 | 0.252 |

## 4. Discussion and Conclusion

Institutional shift persists even with target-tumor overlap in pretraining; fine-tuning 5 WSIs overcomes it, and the Adapter (0.4% of parameters) achieves the best detection F1 on both types, indicating minimal updates suffice under data scarcity. Classification also exceeds all pathologists ($\kappa \approx 0.24$–0.25). **Annotation cost.** Mitotic scoring is non-trivial (36/26 s per HPF without/with AI; Pantanowitz et al., 2020); full-tumor annotation took ∼1 working day/WSI in our experience (informal estimate), so 5 WSIs/type (∼1 pathologist-week) is a one-time per-institution cost that the HITL viewer further amortises via model-suggested candidates. **Limitations:** small test sets (5 cases/type), single institution, H&E-only breast GT.

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
