# OpenReview forum: "Few-Case Fine-Tuning Overcomes Domain Shift in Mitosis Detection and Atypical Classification Across Cancer Types"
_MIDL.io/2026/Short_Papers — MIDL 2026 - Short Papers Poster_

### Official Review · Reviewer_w5Va · 2026-05-03
**Limited experiments but valid and interesting results**

**Rating:** 4
**Confidence:** 4

**Review:**

The results are modest and the experiments limited, but remain valid and are an interesting data-point on the feasibiility and potential of domain adpatation for different applications.

**Summary:**

This paper works on the appication of mitotic figure counting in while-slide-images, focusing on domain shift across institutes. Starting from pre-trained baselines, the authors fine-tune it on "only" 5 new slides, and demonstrate a noticeable improvement both for the detection and classification tasks.

**Strengths:**

- Several baselines and pre-trained foundational models are tested
- Two different applications: detection and classification, are evaluated
- Annotations are performed by two independent pathologists

**Weaknesses:**

- The fine-tuning regimen is not described in great details.
- It is unclear how "costly" it is to annotate even 5 WSI for the domain-adaptation ot be performed. Adding this information would strengthen the paper.

**Justification Of Rating:**

The scope of the experiments are fairly limited, but still provide some value. I would recommend a weak accept ; depending mostly on the acceptance rate the program commitee wishes to attain this year.

---

### Decision · Program_Chairs · 2026-05-08

Accept (Poster)